# Partial Reversible Inhibition of Enzymes and Its Metabolic and Pharmaco-Toxicological Implications

**DOI:** 10.3390/ijms241612973

**Published:** 2023-08-19

**Authors:** Patrick Masson, Aliya R. Mukhametgalieva

**Affiliations:** Biochemical Neuropharmacology Laboratory, Kazan Federal University, 18 ul. Kremlevskaya, 420008 Kazan, Russia

**Keywords:** partial reversible inhibition, hyperbolic inhibition, substrate inhibition, inhibition diagnosis

## Abstract

Partial reversible inhibition of enzymes, also called hyperbolic inhibition, is an uncommon mechanism of reversible inhibition, resulting from a productive enzyme–inhibitor complex. This type of inhibition can involve competitive, mixed, non-competitive and uncompetitive inhibitors. While full reversible inhibitors show linear plots for reciprocal enzyme initial velocity versus inhibitor concentration, partial inhibitors produce hyperbolic plots. Similarly, dose–response curves show residual fractional activity of enzymes at high doses. This article reviews the theory and methods of analysis and discusses the significance of this type of reversible enzyme inhibition in metabolic processes, and its implications in pharmacology and toxicology.

## 1. Introduction

Reversible inhibition of enzymes is a very fast process. Establishment of the enzyme–inhibitor complex occurs within microseconds. With slow-binding inhibitors, establishment of the enzyme–inhibitor steady-state is much longer and can need minutes. To simplify, our topic is devoted to fast reversible inhibition of enzymes (rapid equilibrium assumption), which constitute the majority of encountered reversible inhibition processes. Several mechanisms account for rapid reversible inhibition of enzymes (E). Rapid reversible inhibition can be either competitive, non-competitive, mixed or uncompetitive. In most cases, the enzyme–inhibitor complex (EI) is non-productive. However, in certain cases, reversible inhibition is partial, the ternary complex enzyme–substrate–inhibitor (ESI) can be productive to some extent (Figure 1). The phenomenon of partial reversible inhibition (PRI), rather uncommon, was first theoretically investigated by Cleland [1]. It was called non-linear hyperbolic inhibition because Dixon plots are downward curved in the simplest cases. Shortly after its theoretical prediction, PRI was observed on different purified enzymes: AMP-activated NADH oxidase inhibited by NAD [2], phosphorylase b inhibited by glucose-6-phosphate [3]. In 1975, Irwin H. Segel in his monumental enzyme kinetics textbook [4] thoroughly examined the different cases of reversible inhibition where ESI complexes are productive, leading to partially inhibited enzymes. Segel also derived steady-state rate equations for simple competitive, noncompetitive, mixed-type and uncompetitive inhibition rapid equilibrium enzyme–substrate–inhibitor systems, and even for certain multiple-binding site enzyme models, showing cooperativity or not. Later, Yoshino [5] proposed a convenient graphical method for determination of partial inhibition parameters. Several other graphical methods were also proposed, all of them allow discrimination between the different types of PRI.

The importance of enzyme PRI in metabolic processes has been recognized in early studies [2,3]. The present report deals with formal mechanism analysis of PRI, graphical methods for determination of inhibition parameters, and significant examples of PRI. We also stress the importance of this type of inhibition in toxicology, pharmacology and drug discovery [6]. As a trend, thorough investigations of the mechanisms of action (MOA) of novel drug candidates are becoming a mandatory step in early development of new pharmaceuticals.

## 2. Formal Mechanisms, Analysis and Diagnosis of PRI

Let us consider the rate (*v*) equation (Equation (1)) of the simplest Michaelis–Menten enzyme kinetic system (upper part of Figure 1) with the maximum velocity (*V_max_*) = *k_cat_*[*E*], in which *k_cat_* is the catalytic constant, [*E*] the enzyme concentration, and [*S*] the substrate concentration.
(1)$$v=\frac{k_{cat}\left[E\right]\left[S\right]}{K_{s}+\left[S\right]}$$

A reversible inhibitor (I) may bind to the free enzyme (E) and/or to the enzyme–substrate complex (ES). Four mechanisms account for fast reversible inhibition of enzymes: competitive inhibition, where inhibitor competes with substrate for the same site; non-competitive inhibition; mixed-type inhibition; and uncompetitive inhibition. Uncompetitive inhibitors bind exclusively to ES complexes. In general, the complex ESI is non-productive. However, in the case of PRI, the ESI complex displays some activity (Figure 1 red segment).

Assuming a rapid equilibrium system, Figure 1 shows the minimum mechanism for PRI of a Michaelian enzyme (E). *K_s_* is the dissociation constant of the enzyme–substrate complex, *K_s_* = [*E*][*S*]/[*ES*], and *K_i_* the dissociation constant of the enzyme–inhibitor complex, *K_i_* = [*E*][*I*]/[*EI*]. In this system, the enzyme–substrate–inhibitor complex (ESI) is partially productive with a catalytic constant *βk_cat_* < *k_cat_*.

In Figure 1, the coefficient *α* ≥ 1 (except *α* ≤ 1 for uncompetitive PRI) indicates the change in *K_s_* and *K_i_* resulting from binding of I to the active site. *αK_i_* is sometime written *K’_i_*; the coefficient *β*, 0 < *β* < 1, indicates that ESI complex is productive. Then, in the case of competitive inhibition, *α* = ∞ and *K_i_* << *αK_i_*; for non-competitive inhibition, *α* = 1 and *K_i_* = *αK_i_*; for uncompetitive inhibition, *K_i_* = ∞ and *αK_i_* << *K_i_*. In the case of mixed-type inhibitions, competitive mixed-type, *α* > 1 and *K_i_* < *αK_i_*; uncompetitive mixed-type, *α* < 1 and *αK_i_* << *K_i_.*

Assuming rapid equilibrium, the general scheme (Figure 1) can be described by the following rate equation (Equation (2)):(2)$$v_{i}=\frac{\left[\left(k_{cat}\left[E\right]\left[S\right]\right)/K_{s}\right]+\left[\left(\beta k_{cat}\left[E\right]\left[S\right]\left[I\right]\right)/\left(\alpha K_{s}K_{i}\right)\right]}{1+\left(\left[S\right]/K_{s}\right)+\left(\left[I\right]/K_{i}\right)+\left(\left(\left[S\right]\left[I\right]\right)/\left(\alpha K_{s}K_{i}\right)\right)}$$

At high inhibitor concentration, [*I*], the maximum velocity at saturating substrate concentration tends to *V_max,i_* = *βk_cat_*[*E*]. At first, Lineweaver–Burk plots (1/*v* vs. 1/[*S*]) and hyperbolic replots of slope and 1/*v*-axis intercept as a function of inhibitor concentration [*I*] can conveniently be used for diagnosis of PRI. 3D representations of Lineweaver–Burk, Hanes and Eadie-Hofstee plots at different inhibitor concentrations can be built [7]. Three-dimensional graphical representations are theoretically appealing. However, experimental errors, in particular at low substrate concentrations, can make it difficult to interpret 3D diagrams and to provide an unambiguous diagnosis. Non-linear regression fitting of Equation (2) can theoretically be performed to determine parameters of PRI. This applies to simple Michaelis–Menten models. However, such a graphical method may need a large number of experimental points if the catalytic scheme involves several inhibitor binding sites. In these cases, the saturation curve is not a simple monophase hyperbole. This issue will be emphasized in the next paragraphs.

Several other graphical methods allow discrimination between full inhibition and PRI and lead to the determination of parameters of PRI. In particular, let us mention the specific velocity (*v*_0_*/v_i_*) plot of Baici [8], *v*_0_*/v_i_* vs. *σ*/(1 + *σ*) where *σ* = [*S*]/*K_m_*. More recently, a decrease in secondary plots of *^app^K_m_/^app^k_cat_* vs. [*I*] was found to be a valuable qualitative criterion for partial inhibition [9]. About this later plot, although the authors insist that analysis of data must be performed using non-linear regression, we must point out that propagation of errors on *^app^K_m_* and *^app^k_cat_* parameters may impair accurate determination of *β* values. However, simpler, more convenient and accurate plots can be used. The following describes these graphical methods. 

According to Copeland [10] (pp. 336–338), direct concentration–response plot, i.e., the fractional enzyme activity, *y* = *v_i_/v*_0_, as a function of [*I*] can be conveniently drawn, using *IC*_50_ as the medium inhibitory concentration (Equation (3), Figure 1). At [*I*] = 0, *y* = 1 and at high [*I*], the residual fractional activity is *y_i,min_*. In Equation (3), *IC*_50_ is related to inhibition constant *K_i_*.
(3)$$\frac{v_{i}}{v_{0}}=\frac{1-y_{i,min}}{\left[1+\left(\frac{\left[I\right]}{{IC}_{50}}\right)\right]}+y_{i,min}$$
here *v*_0_ is the rate at given substrate concentration in the absence of an inhibitor.

For full reversible inhibitors, relationships between *K_i_* and *IC*_50_ were first established by Cheng and Prusoff [11]. A recent review presents relationships between *K*_i_ and indicators of pharmacological/toxicological activity, see [12]. In the case of PRI, relationships between *K_i_* and *IC*_50_ are developed for each inhibition mechanism have been established by Gelpi et al. [13]. The authors pointed out that several *IC*_50_ relationships are identical between full and partial reversible inhibitors. Moreover, in the case of PRI, the meaning of *IC*_50_ has to be reconsidered. Indeed, depending on the value of *β*, the real *IC*_50_ value for a PRI is less than the value for a full inhibitor that inhibits 50% of the total enzyme activity. Also, from toxicological and therapeutic points of view, in the case of PRI, if *IC*_50_ values were experimentally determined from *v_i_*/*v*_0_ plots, it would be more relevant to define these values as apparent *IC*_50_ and to provide the value of *β.* Therefore, the sole determination of *IC*_50_ cannot be used for diagnosis of PRI. Only the residual value of *v_i_/v*_0_ at high [*I*] provides evidence for PRI (Equation (3), Figure 1). 

Then, Dixon plots can be built. In steady-state kinetics, deviations from linearity in Dixon plot (1/*v_i_* versus [*I*] at different substrate concentrations) [14] or reversible enzyme inhibition may provide a first indication in favor of partial inhibition (cf., [10]). Downward curvature of Dixon plots as inhibitor concentration increases may indeed reflect productivity of the ternary complexes ESI. However, quantitative determination of PRI parameters is difficult from Dixon plots (cf. Equation (4)):(4)$$\frac{1}{v}=\frac{1}{V_{max}}\left[\frac{\left(1+\frac{\left[I\right]}{\alpha K_{i}}\right)}{\left(1+\frac{\beta \left[I\right]}{\alpha K_{i}}\right)}\right]+\frac{K_{s}}{V_{max}\left[S\right]}\left[\frac{\left(1+\frac{\left[I\right]}{K_{i}}\right)}{\left(1+\frac{\beta \left[I\right]}{\alpha K_{i}}\right)}\right]$$

On the other hand, upward curved Dixon plots, indicate multisite inhibitor interactions. 

More complex Dixon plots can be observed. Multiphasic Dixon plots with consecutive downward and upward curvatures can be seen in the case of the binding of a second inhibitor molecule on an allosterically distinct site affects the coefficient *β*, leading to coefficient *δ* with *δ* < *β*. The inhibition of human BChE-catalyzed hydrolysis of phenylacetate (reporter substrate) by acetylthiocholine (competing substrate) is described by such a mechanism (Figure 2, Figure 2) [15].

Figure 2 deals with the complex inhibition pattern of human ChEs as seen in Figure 2. Inhibition path is first competitive inhibition at low inhibitor concentration followed by two consecutive partial inhibition steps as inhibitor concentration is increased. I_p_ refers to binding of a second inhibitor molecule on a peripheral (p) site that induces an allosteric effect on the catalytic active center. The two sites are interconnected through a highly mobile loop, called Ω loop. Both complexes ESI and I_p_ESI are partially productive (*δ* < *β* < 1). For theoretical analysis and derivation of formal equations, see supplementary information in [15]. 

As mentioned above, Lineweaver–Burk plot analysis (1/*v* versus 1/[*S*] at different [*I*]) can be used for PRI diagnosis. It has also been proposed for determination of PRI parameters. This approach was extensively developed in Segel’s book [4] and in [13]. In particular, in their thorough theoretical approach, these authors examined all possible situations with respect to all cases of reversible inhibitions (competitive, non-competitive, mixed and uncompetitive) for discrimination between full and partial reversible inhibitors. Replots of 1/*v_i_* and slope versus [*I*] are hyperbolic for PRI, while they are linear for fully reversible inhibition. On the other hand, replots of 1/Δ intercepts (on x and y axes) and 1/Δ slope as a function of 1/[*I*] are linear for PRI and provide *K_i_*, α and β. (Δ values are experimental values of intercepts or slopes at concentration [*I*] minus experimental values of intercept or slope in the absence of an inhibitor).

Kinetic analysis based on the integration of the Michalis–Menten equation, i.e., analysis of the time course of reporter substrate degradation until substrate consumption, in the presence of an inhibitor, can also provide evidence for partial inhibition. The modern theoretical analysis of competing substrate progress curves kinetics was developed by Golicnik and Masson [16]. Such an analysis implies that initial rates in the presence and absence of a competitor, *v_i_* and *v*_0_, respectively, have to be determined at the beginning of progress curves, immediately after rapid mixing of a reaction mixture. The concentration of the reporter substrate [S] is low, much less than *K_m_*, thus *v*_0_ = (*k_cat_/K_m_*)[E][S]. In the case of fully competitive inhibition where *K_m,app_* = *K_m_*(1+[*I*]/*K_i_*), *v_i_/v*_0_→0 at high[*I*]. Thus, the rate equation (Equation (2)) simplifies and in the case of PRI, at low [S], the hyperbolic plot of *v_i_/v*_0_ vs. competitor concentration does not reach 0 activity at high competing substrate concentration (i.e., high [*I*]): (5a)$$\frac{v_{i}}{v_{o}}=\frac{1+\beta \frac{\left[I\right]}{{\alpha K}_{i}}}{\left(1+\frac{\left[I\right]}{K_{i}}\right)}=\frac{\frac{1}{\left[I\right]}+\frac{\beta }{{\alpha K}_{i}}}{\frac{1}{\left[I\right]}+\frac{1}{K_{i}}}$$

Rather, when [*I*] become high (>>*K_i_*), 1/[*I*]→0, and *v_i_/v*_0_ reaches an asymptotic value depending on *β* and *αK_i_* [15,17]. In the simplest case where *α* = 1, the asymptotic limit is *β* (Equation (5a)). It is interesting to note that Equation (5a) is analogous to Equation (3), in particular in the case of competitive inhibition where *IC*_50_ = *K_i_*.

*K*_i_ and *β* can be determined by non-linear fitting of Equation (5a). If *α* ≠ *1,* then *v_i_*/*v*_0_→*β*/α at high [*I*]. In the case of multiple binding sites for inhibitors, causing PRI with coefficients *β* and *δ,* Equation (5a) becomes Equation (5b): (5b)$$\frac{v_{i}}{v_{0}}=\frac{\left(1+\frac{\beta \left[I\right]}{{\alpha K}_{i}}+\frac{\delta {\left[I\right]}^{2}}{{\alpha K}_{i}{\gamma K}_{ip}}\right)}{\left(1+\frac{\left[I\right]}{K_{i}}+\frac{{\left[I\right]}^{2}}{K_{i}K_{ip}}\right)}$$
with *v_i_*/*v*_0_→δ⁄αγ at [*I*] >> *K_ip_*. This situation is described in Figure 2. In that case, all graphical representations are not monophasic (cf. Figure 2 and supplementary information in [15]). Therefore, thorough analysis of PRI needs a large number of experimental points. 

Figure 3 shows an example of partial inhibition of human BChE (*β* = 0.18) determined from the time course of hydrolysis of benzoylthiocholine (reporter substrate) in the presence of varying concentrations of a competing substrate, benzoylcholine (blind competing substrate). If the blind substrate were a fully competing substrate of benzoylthiocholine, BChE-catalyzed hydrolysis, *v_i_*/*v*_0_ should reach 0 at a high benzoylcholine concentration. The asymptotic limit (*β*) indicates partial inhibition. 

More conveniently, the coefficient *β*, (0 < *β* < 1), can be simply determined from linear plots of fractional velocity (*v_i_*/(*v*_0_
*− v_i_*)) as a function of the reciprocal of [*I*] at different [*S*] (Equations (6)–(9); Table 1). This graphical method was developed by Yoshino [5] and popularized by Whiteley [19]. The general equation corresponding to the system described in Figure 1 applies to mixed-type inhibition and can be expressed by (Equation (6a)). Because *β* > 0, at 1/[*I*] = 0, the ordinate intercepts values are always positive, depending on [*S*], *K_s_*, *α* and *β* (Equation (6b)):(6a)$$\frac{v_{i}}{v_{0}-v_{i}}=\frac{\left(1+\frac{\left[S\right]}{K_{s}}\right)\left(\frac{\alpha K_{i}}{\left[I\right]}+\beta \right)}{\left(1-\beta \right)\left(\frac{\left[S\right]}{K_{s}}\right)+\left(\alpha -\beta \right)}$$
(6b)$$\frac{v_{i}}{v_{0}-v_{i}}=\frac{\left(1+\frac{\left[S\right]}{K_{s}}\right)\beta }{\left(1-\beta \right)\left(\frac{\left[S\right]}{K_{s}}\right)+\left(\alpha -\beta \right)}$$
(6c)$$\mathrm{and}\;\mathrm{the}\;\mathrm{slope}=\frac{\left(1+\frac{\left[S\right]}{K_{s}}\right)\alpha K_{i}}{\left(1-\beta \right)\left(\frac{\left[S\right]}{K_{s}}\right)+\left(\alpha -\beta \right)}$$

Then, equations for competitive, non-competitive and uncompetitive full and partial inhibitions are as follows: 

For partial competitive inhibition (*β* = 1, *α* > 1):(7a)$$\frac{v_{i}}{v_{0}-v_{i}}=\frac{\left(1+\frac{\left[S\right]}{K_{s}}\right)}{\left(\frac{\alpha -1}{\alpha K_{i}}\right)}\left(\frac{1}{\left[I\right]}+\frac{1}{\alpha K_{i}}\right)$$
(7b)$$\mathrm{with}\;\mathrm{ordinate}\;\mathrm{intercept}=\frac{\left(1+\frac{\left[S\right]}{K_{s}}\right)}{\left(\alpha -1\right)}$$
(7c)$$\mathrm{slope}=\frac{\left(1+\frac{\left[S\right]}{K_{s}}\right)}{\left(\frac{\alpha -1}{\alpha K_{i}}\right)}$$
and intercept on abscissa is given in Table 1. 

For partial non-competitive inhibition: (8)$$\frac{v_{i}}{v_{0}-v_{i}}=\frac{1}{1-\beta }\left(\frac{K_{i}}{\left[I\right]}+\beta \right)$$

Ordinate intercept and slope are reported in Table 1. 

For partial uncompetitive inhibition: (9a)$$\frac{v_{i}}{v_{0}-v_{i}}=\frac{\left(1+\frac{\left[S\right]}{K_{s}}\right)}{\left(1-\beta \right)\left(\frac{\left[S\right]}{K_{s}}\right)-\beta }\left(\frac{\alpha K_{i}}{\left[I\right]}+\beta \right)$$
(9b)$$\mathrm{Ordinate}\;\mathrm{intercept}=\frac{\left(1+\frac{\left[S\right]}{K_{s}}\right)\beta }{\left(1-\beta \right)\left(\frac{\left[S\right]}{K_{s}}\right)-\beta }$$
(9c)$$\mathrm{Slope}=\frac{\left(1+\frac{\left[S\right]}{K_{s}}\right)\alpha K_{i}}{\left(1-\beta \right)\left(\frac{\left[S\right]}{K_{s}}\right)-\beta }$$
and replot of slope vs. [*S*] is a hyperbolic function with asymptote *αK_i_*/(1 − *β*) at high [*S*].

For full inhibition, intercepts on abscissa and ordinate are equal to 0 for all type of inhibitors.

In the particular case of partial substrate inhibition (i.e., partial inhibition by excess substrate), the model in Figure 1 can be used with *S* instead of *I*. Thus, Yoshino [20] proposed a simple graphical method for determination of substrate inhibition constants (*K_si_* and *αK_si_*) and coefficient *β*. The method consists of plotting *v*/(*V_max_* − *v*) as a function of the reciprocal of substrate concentrations. The intercept on the *y*-axis is *β*/(1 − *β*).

Finally, in a recent master article, Grant [21], discussed the graphical analysis of PRI, showing all types of plots used for discrimination between PRI (Figure 1) and other types of conventional and unconventional fast reversible inhibition of simple enzyme systems. Moreover, the author showed that these graphical approaches can be used for identification and quantification of the different mechanisms of non-essential enzyme activation where *β* > 1. This includes uncommon cases of inhibitors causing apparent activation at low substrate concentration and inhibition at high substrate concentration. This very rare situation with *α* < 1 and where the inhibitor acts as an activator (*β* > *α*) at low substrate concentration was called partial inhibition system C5 by Segel [4]. In such a case, the Lineweaver–Burk plots intersect in the upper right quarter [15,22]. With this particular case, we need to draw attention to the problem of non-essential activation of enzymes displaying a mechanism similar to the one described in Figure 1, but with *β* > 1. Such systems involving an activator (A) can be analyzed like PRI, with *K_a_*, the activation constant instead of *K_i_* and apparent catalytic parameters are (Equations (10) and (11)):(10)$$V_{max,a}=\frac{V_{max}\left(1+\frac{\beta \left[A\right]}{\alpha K_{a}}\right)}{\left(1+\frac{\left[A\right]}{\alpha K_{a}}\right)}$$
(11)$$K_{s,a}=K_{s}\frac{\left(1+\frac{\left[A\right]}{K_{a}}\right)}{\left(1+\frac{\left[A\right]}{\alpha K_{a}}\right)}$$

All these methods apply to monomeric and oligomeric enzymes showing no cooperativity in substrate/inhibitor binding as well as to monomeric enzymes displaying allosteric PRI. For more complex enzyme systems, including multi-subunit allosteric enzymes, the Segel book [4] remains the essential reference. 

To check the validity of the PRI mechanistic model and parameters determined by non-linear or linear regression analysis of steady-state inhibition data, statistical analysis of residuals can be used. This statistical method popularized by Cornish-Bowden [23] is based on comparison of the distribution of residual sum of squares (*Q*^2^) between different alternative mechanistic models. It allows discrimination between different steady-state kinetic models. 

For a complete mechanistic description of PRI, we must point out that the classical kinetic approach should be completed by in silico methods, i.e., docking of inhibitors and mechanistic consequence on catalysis by quantum mechanics/molecular mechanics (QM/MM) simulations. 

## 3. Possible Experimental Artefacts

Limited solubility of inhibitors beyond a certain concentration can be the cause of pseudo-partial inhibition. Thus, analytical measurement of inhibitor concentration in buffer solution, and the use of co-solvent to increase inhibitor solubility must be performed to eliminate this possible artefact. Allosteric inhibition and product inhibition (i.e., non-dissociation of reaction products from the enzyme active center) may also cause pseudo-partial inhibition. Thus, primary plots (e.g., dose-response curves), Hill plots and secondary plots (e.g., Dixon plots, replots of 1/*v_i_* and slope versus [*I*] from Lineweaver–Burk plots) must be systematically built to avoid misdiagnosis of partial inhibition.

## 4. Relevance of PRI

### 4.1. Metabolic Processes

Metabolic processes can be modulated or corrected by the action of PRI on a key enzyme reaction. The simplest and quite common case is partial inhibition of an enzyme by excess substrate [20]. Inhibition by excess substrate is rather common and plays important roles in regulation of metabolic processes and physiological systems [24]. This type of inhibition was first analyzed by Haldane who considered formation of ternary complex SES at high substrate concentrations. In the Haldane model, SES is partially productive. However, numerous mechanistic models account for substrate inhibition [25]. Several examples illustrate the physiological importance of partial substrate inhibition.

Cholinesterases, acetylcholinesterase (AChE) and butyrylcholinesterase (BChE), important enzymes in the central and peripheral cholinergic system, deviate from the Michaelis–Menten model with positively charged substrates. In particular, AChE is inhibited by excess acetylcholine, the cholinergic system neurotransmitter, while the sister enzyme BChE is activated by excess substrate [15,26]. In central cholinergic synapses and at neuromuscular junctions where both enzymes coexist, this property may serve as a first line of defense to protect the cholinergic system in response to a rapid increase in the acetylcholine level, e.g., in case of acute inhibition of AChE by a toxicant. Thus, BChE has been regarded as a surrogate or a backup of AChE.

To maintain constant glucose levels in the body, there is regulation of the activity of a multi-enzyme complex, the pyruvate dehydrogenase complex, which is inhibited at high concentrations of substrates: pyruvate, ADP, NAD and CoA [27]. Many physiological phenomena are controlled by substrate inhibition. According to [28] about a quarter of the known substrates have such an inhibiting property. Using the example of a mutation in the haloalkane dehalogenase LinB from *Sphingobium japonicum* UT26, it was shown that strong enzyme inhibition by substrate occurs. This inhibition was attributed to direct blockage and/or limited conformational flexibility of the product exit from the active site of the enzyme.

Reed et al. [24] listed more than 60 enzymes showing partial substrate inhibition. Now the number of enzymes displaying PRI is likely much higher. In particular, several systems are clearly of metabolic, physiological, and epigenetic significance. For example, inhibition of phosphofructokinase by ATP acts in regulation of glycolysis; inhibition of tyrosine hydroxylase by tyrosine and inhibition of tryptophan hydroxylase by tryptophane play a role in homeostasis of dopamine synthesis; inhibition of the folate cycle enzymes by folate plays a role in folate homeostasis, and because of seasonal variations in the content of folate and B vitamins of diets, PRI of the folate cycle enzymes maybe regarded as an evolutionary mechanism to dampen the large seasonal fluctuations in diet folate availability; inhibition of DNA methyltransferase-1 by regions of unmethylated DNA protect these regions from inappropriate pathogenic methylations. Lastly, it was shown that partial inhibition of glycolysis led to inhibition of atherosclerotic plaque growth and caused a positive effect on cardiac function [29].

### 4.2. Pharmacology and Toxicology

In addition to the previous example about inhibition of AChE by excess acetylcholine, we must consider the inhibition of ChEs by various toxic compounds. First, it is important to make a difference between the irreversible and reversible inhibitors of these enzymes. Irreversible inhibition of ChEs happens in acute poisoning by anti-cholinesterase compounds, e.g., carbamates and organophosphates, mostly pesticides and banned chemical warfare agents [30]. These compounds react with the enzyme catalytic serine. Thus, they are irreversible or quasi-irreversible inhibitors. These types of inhibitors highlight the toxicological and pharmacological importance of irreversible inhibition of cholinesterases. Carbamylation and phosphylation (phosphorylation and phosphonylation) of AChE by carbamates and organophosphates is a major public health concern that causes the death of more than 100,000 people annually around the world [30]. Cholinesterases can also be reversibly inhibited by various ligands. Several works pointed out that different types of ligands may cause partial inhibition of these enzymes. These molecules can be either toxicants such as d-tubocurarine or drugs of interest in the palliative treatment of Alzheimer disease or other pathological states [15,22,31]. For inhibition of cholinesterases, the mechanism of PRI involves the binding of inhibitors on peripheral binding sites, in particular to the peripheral anionic site (PAS) at the entrance of the active site gorge. This binding triggers the partial inhibition of acylation by positively charged substrates (e.g., the natural substrate, acetylcholine) via an allosteric effect, cf. Figure 2. This mechanism has been established from in vitro studies. Thus, in vivo partial inhibition of the activities (natural or promiscuous) of cholinesterases by various ligands of cholinergic receptors or competing substrates (cf. Figure 3) cannot be ruled out and may be pharmacologically and toxicologically relevant. For example, a series of tetracyclic thienopyrimidines synthesized as possible candidate drugs against Alzheimer’s disease (AD) were found to display uncompetitive PRI of AChE with *β* up to 0.24 [32]. Recently synthesized indanone derivatives as multifunctional inhibitors of ChEs for AD are protective of neuronal cells against hydrogen peroxide and against aggregation of Aβ_1–40_ aggregation. These compounds are also partial non-competitive inhibitors of both ChEs [33]. The partial inhibition of ChEs by these compounds may indeed result from an allosteric effect due to the binding of inhibitors on the enzyme peripheral anionic site, but it is not yet known whether this property has potential consequences for the pharmacological effectiveness of these compounds on AD animal models. In the case of PRI, because the residual enzyme activity (*v_i_/v*_0_) tends towards *β* (cf. Equations (2) and (3) and Figure 1 and Figure 3) at high concentrations of inhibitor (high doses or “overdoses” of a drug), PRI can be regarded as a protective mechanism to limit the deleterious physiological effects of a reversible toxicant on a key metabolic enzyme. The following examples support this hypothesis.

Trimetazidine is an effective therapeutic drug for the treatment of various forms of coronary heart diseases. The metabolic effects of trimetazidine, used in the clinic, increase the efficiency of cardiac work with partial inhibition of β-oxidation of fatty acids, allowing hydrolysis of ATP to be more efficiently converted into contractile heart work [34]. Additionally, partial inhibition is observed for cytochromes P450. These enzymes play an important role in the oxidation of numerous endogenous and exogenous compounds in the body. The P450 family of enzymes can metabolize several substrates simultaneously because they bind to different regions of the active center. For example, it was shown that the simultaneous action of testosterone and erythromycin/midazolam/terfenadine, causes incomplete inhibition of the metabolism of the other substrates even at saturating concentrations [35]. Thus, PRI in drug metabolism mediated by cytochromes P450 can have significant pharmacological or toxicological consequences. The following example highlights this statement. Partial inhibition of the Gyps vulture cytochrome P450 (CYP2C9) by the non-steroidal anti-inflammatory drug diclofenac was inferred indirectly from pharmacokinetic profiles, i.e., longer mean residence time (MRT) and *t*_1/2_ for the most sensitive birds [36]. These parameters were correlated with *LD*_50_ values. Thus, the increased sensitivity to diclofenac of the Gyps vulture compared to other related birds and chickens may be of pharmacogenomic origin. However, in other situations, increased toxicity due to partial inhibition of a detoxifying enzyme can have non-genetic origins, e.g., dysfunction of a metabolic chain, multiple antagonistic interactions on target enzymes.

Partial inhibition of enzymes may confer greater safety in the use of effective but dangerous drugs in maintaining a low level of enzyme activity. A good example is given by the partial allosteric inhibition of thrombin by sulfated coumarins [37]. These compounds apparently inhibit 50% of thrombin at saturation with a lower apparent *IC*_50_ compared to orthosteric inhibitors (Figure 4). Thus, they retain the anticoagulant property but reduce the risk of bleeding. This approach extended to partial inhibition and regulation of other coagulation factors as well as to regulatory allosteric enzymes may have important therapeutical implications.

On the contrary, PRI of enzymes may limit the action of inhibitors and therefore their pharmacological interest. For instance, melanin formation in skin and brain from naturally occurring monophenols depends on tyrosinase. Inhibitors of this enzyme have been developed to treat hyperpigmentation of the skin but could be of interest in treatment of neurodegenerative diseases. It was found that certain benzonitrile derivatives of mushroom benzonitriles are PRI of tyrosinase (*β* up to 0.42) [38]. The high value of *β* may limit the therapeutic interest of such inhibitors.

PRI of enzymes involved in the metabolism of drugs, may affect clinical drug interactions. An example is given by the partial (*β* = 0.12) mixed-type inhibition of UDP-glucuronosyl transferase isoenzymes involved in paracetamol (acetaminophen) glucuronidation caused by dasabuvir, an anti-hepatitis C drug [39]. Correlation between in vitro and pharmacokinetic data makes it possible to predict the clinical interactions of both drugs. Then, in the case of the co-administration of both drugs to patients, to avoid the potential hepatotoxicity of paracetamol, it is recommended to reduce the dose of paracetamol. 

The typical antipsychotic drug risperidone, used for the treatment of schizophrenia, has been reported to cause hyperbolic inhibition of the uncompetitive type on D-amino acid oxidase with an inhibition constant in the low micromolar range [40]. Because in vitro and mathematical and computer modeling studies indicate that uncompetitive inhibitors are more effective than other reversible inhibitors in open systems [41], this suggests that such partial uncompetitive inhibitors may have a good prospect for obtaining more effective therapeutic effects. This could be of great interest in the treatment of diseases associated with hyper- and hypofunction of neurotransmission but also in numerous enzyme inhibitor-based therapies.

Another potent practical therapeutic application could be to optimize the combination of synergistic drugs acting as enzyme inhibitors within large metabolic networks, and causing the minimal side effects. Current optimization approaches imply modulation of the dosage of different drugs for reducing the activity of target enzymes. Complex algorithms have been proposed for solving this problem [42]. Alternatively, a selection of drugs acting as partial inhibitors of key enzymes would ensure a low level of activity of selected target enzymes. For this purpose, research of partial inhibitors of such enzymes is a prerequisite.

## 5. Conclusions 

Partial inhibition of enzymes by ligands and an excess of natural or competing substrates is a quite common phenomenon. However, PRI has been insufficiently explored, analyzed, and exploited so far. Several simple graphical methods make it possible to detect PRI and to determine the inhibition parameters, in particular the coefficient *β*. The relevance of enzyme PRI in physiological processes is well documented where it is involved in multiple regulatory metabolic functions. Participation of PRI in epigenetic mechanisms is still poorly documented. In toxicology, PRI of target enzymes may dampen the acute action of toxicants and participate in defense mechanisms against poisonous natural and artificial xenobiotics. In pharmacology, enzyme inhibitors acting as PRI show certain therapeutic advantages over classical enzyme inhibitors in use in medicinal treatments. This is in particular the case with partially reversible uncompetitive inhibitors. Therefore, in the early stages of drug discovery [6], careful investigation of PRI must enter in the arsenal of mechanistic tools for investigating enzyme inhibition. In particular, new quantitative structure–activity relationship (QSAR) approaches [12] and drug design methods must take into account this type of inhibition. As we pointed out, the classical enzyme inhibition calculation methods we developed for determination of PRI parameters must be associated to in silico approaches for a complete mechanistic description of PRI, and in particular to determine how complexes ESI are productive. Conversely, modification of the inhibitory properties of therapeutic enzymes by switching their behavior from full inhibition to PRI can be considered by computer design of mutations or directed evolution. In the future, this should help to optimize the activity of novel mutated enzymes of medical interest.

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
