# Peer review of "Partial Reversible Inhibition of Enzymes and Its Metabolic and Pharmaco-Toxicological Implications"

_ijms, 2023, doi:10.3390/ijms241612973_

Round 1

Reviewer 1 Report

The review by Massonet al. introduces the concept and recent advances in partial reversible inhibition of target proteins. This mechanism holds significant importance in the early discovery and development of drug candidates. Moreover, this mechanism paves the way for further structure-activity relationship (SAR) studies, offering potential for deeper insights into drug design. The authors present a in-depth analysis of the formal mechanisms involved in partial reversible inhibition, highlighting its close association with metabolic diseases. Overall, this review provides valuable insights into the concept of partial reversible inhibition and its application in drug discovery. However, in order to enhance the manuscript, I believe there are some concerns that the authors should address.

1.      In the introduction part, the author should introduce the concept and importance of the mechanism of action (MOA) study in the drug discovery, introduce three main types of inhibition, and then introduce the concept of partial reversible inhibition, show their differences.

2.      In the “formal mechanisms, analysis and diagnosis of PRI” part, the authors give different equations, which is impressive. But it’s also difficulty to digest for readers. Is it possible the authors explain more by using real interaction between drugs and target enzymes.

3.      The third part of this review, experimental artefact, there is only several sentences and no references are cited, which is odd for review paper.

4.      In the “pharmacology and toxicology” part, the authors highlighted the inhibition of AChE and cytochromes P450 through partial reversible mechanism. There are more than 60 enzymes can be inhibited through partial substrate inhibition. I hope the author could give more detailed examples. More importantly, figures should be provided to illustrate the mechanism of drugs examples used in this part.

5.      It would be great if the authors could give an outlook and forecast future research directions in this filed.

Author Response

Answers to reviewer 1

Changes in the text are in red.

We revised the manuscript in compliance with all comments. We are grateful to comments and recommendations. In particular the different types and characteristics of reversible inhibitors were developed (line 63-68; 80-84). Future developments of PRI methodology and perspectives were mentioned (lines 245-248) and in conclusion (lines 415-424).

However, about item 3 (artefacts), we are unable to provide references. What we wrote is just from our past experience, but we guess that most enzymologists were facing such problems (substrate solubility, inhibition by reaction products) and are aware how to solve them. About item 4, more examples can be found in ref 22 (Reed et al). We did not want to repeat the examples these authors published, and thus provided the most significant cases published since this paper, in particular our recent works on cholinesterases (ref 15-17).

Patrick Masson

Author Response

Answers to reviewer 2

Changes in the text are in green.

We revised the manuscript in compliance with all comments. We are grateful to comments and recommendations.

  1. Ref 6 is correct
  2. About Fig 2, a sentence was added in the text (line 140-143) and Fig 2 legend slightly modified.
  3. The evolutionary mechanism was mentionaed (lines 292-295)
  4. The sentence was rephrased and splitted in several parts, lines 300-307.
  5. Phosphylation is correct, this is the general term for phosphorylation or phosphon
  6. Gyps vultures: we corrected, and added the case of chicken that clarifies the sentence.
  7. The meaning of IC50 in PRI is very important. Therefore we added several sentences to discuss this issue in section 2 (lines 119-123) and in the example of anticoagulants, we wrote apparent IC50 (line 360)
  8. Tyrosinase and its inhibitors, corrected (lines 374-375).

Patrick Masson